# A New Role for Yeast Cells in Health and Nutrition: Antioxidant Power Assessment

**DOI:** 10.3390/ijms241411800

**Published:** 2023-07-22

**Authors:** Thomas Gosselin-Monplaisir, Adilya Dagkesamanskaya, Mylène Rigal, Aurélie Floch, Christophe Furger, Hélène Martin-Yken

**Affiliations:** 1TBI, Université de Toulouse, CNRS, INRAE, INSA, 31400 Toulouse, France; monplaisir.thomas@gmail.com (T.G.-M.); dagkesam@insa-toulouse.fr (A.D.); 2Anti Oxidant Power AOP, 31000 Toulouse, France; mrigal@laas.fr (M.R.); afloch@laas.fr (A.F.); cfurger@laas.fr (C.F.)

**Keywords:** antioxidant, biosensor, carotenoids, LUCS, synthetic biology, yeasts

## Abstract

As the use of antioxidant compounds in the domains of health, nutrition and well-being is exponentially rising, there is an urgent need to quantify antioxidant power quickly and easily, ideally within living cells. We developed an Anti Oxidant Power in Yeast (AOPY) assay which allows for the quantitative measurement of the Reactive Oxygen Species (ROS) and free-radical scavenging effects of various molecules in a high-throughput compatible format. Key parameters for *Saccharomyces cerevisiae* were investigated, and the optimal values were determined for each of them. The cell density in the reaction mixture was fixed at 0.6; the concentration of the fluorescent biosensor (TO) was found to be optimal at 64 µM, and the strongest response was observed for exponentially growing cells. Our optimized procedure allows accurate quantification of the antioxidant effect in yeast of well-known antioxidant molecules: resveratrol, epigallocatechin gallate, quercetin and astaxanthin added in the culture medium. Moreover, using a genetically engineered carotenoid-producing yeast strain, we realized the proof of concept of the usefulness of this new assay to measure the amount of β-carotene directly inside living cells, without the need for cell lysis and purification.

## 1. Introduction

Often used as dietary supplements and preservatives in food and feed products, antioxidants play a key role in health and nutrition where high-nutritional-grade products are required. While there are a lot of synthetic antioxidants available, the demand for natural antioxidants is very high. In this context, their production in microorganisms as “microbial cell factories” is green and sustainable from both environmental and economic standpoints. There is a strong worldwide competition for the production of antioxidants in microorganisms accompanied by a need to follow their synthesis during the production processes. Finally, for the synthetic biology field, the idea to monitor the biosynthesis of compounds directly within the producing microorganisms (e.g., bacteria, yeasts, micro-algae, etc.) is extremely appealing and constitutes a goal in itself. Several groups have attempted to develop ways to control and monitor this synthesis, which would ideally require an efficient detection of the antioxidant compounds as they are produced. Different approaches using either in vitro conditions, living cells, animal models or human cohorts (for clinical trials) are traditionally used to address the antioxidant properties and/or effects of various natural and synthetic substances. One of those approaches, the AOP1 assay proposed by Furger and coworkers, is based on a technology called a Light-Up Cell System (LUCS) originally developed on mammalian cells to monitor the homeostasis status [1]. This method uses the photoinduction of the cyanine dye Thiazole Orange (TO) to induce and control intracellular ROS (reactive oxygen species) production, leading to fluorescence variations. In the LUCS method, cell homeostasis/viability can then be assessed by comparing fluorescence measurements before and after the photoinduction of TO by light application. Using a modified version of the lighting sequence led to the development and validation of AOP1, a live cell assay dedicated to the antioxidant assessment of standard compounds [2] and a wide range of vegetal extracts [3,4] entering the cell and acting as free-radical scavengers.

In this work, we developed a new assay for antioxidant power detection based on the yeast *S. cerevisiae* as a cell model and using TO photoinduction to produce ROS intracellularly. Yeast was chosen as the preferred candidate for multiple reasons: easy and cheap culturing, manipulation and storage, the existence of advanced genetic tools and the possibility to test the effect of internally produced antioxidant molecules, since many different synthetic metabolic pathways have been developed in yeasts (for reviews, see [5,6,7,8]).

## 2. Results

### 2.1. AOPY: A Transposition of the AOP1 Live Cell Assay to Yeast

The AOPY (Anti Oxidant Power in Yeasts) process is an adaptation of the AOP1 assay developed and currently used on mammalian cells [2]. This assay is based on the LUCS technology [1], and its principle is simple: live cells are incubated with a fluorescent cyanine dye (TO) until an equilibrium plateau is attained. Then, cells are subjected to light pulses which photo-activate the TO molecule resulting in the production of intracellular ROS followed by an increase in TO fluorescence. To set up the AOP1 technology on the yeast system, we started by choosing a widely used yeast model: *Saccharomyces cerevisiae BY4741*. Then, we defined the test parameters suitable for this biological model. The growth phase of the yeast culture, cell density and the concentration of the TO for the assay were all optimized. Typically, yeast cells were incubated for 30 min in TO containing the reaction mixture before the illumination (470 nm). The increase in the TO fluorescence after the light flashes, measured at 535 nm, was the main parameter indicating the efficiency of the test in the studied conditions. 

#### 2.1.1. Adjusting the Yeast Cell Density Parameter

In order to define optimal cell density for the measurements, we compared the illumination results in the presence of TO on the serial dilutions of a yeast cell culture in the exponential growth phase from OD_600_ 1 to 0.3. TO concentration was fixed for these experiments at 64 µM. We observe that, in the same way as for higher eukaryotic cells, the TO fluorescence signal increases with the application of light flashes validating the transposition of the AOP1 technology to *S. cerevisiae*. The highest ratio of the post/pre-illumination fluorescence plateau (Fpost/Fpre) was observed for yeast at OD_600_ = 0.6 (see Figure 1b), which indicates that this cell density is likely to provide the optimal condition for the test.

#### 2.1.2. Adjusting the Thiazole Orange Concentration

The second parameter to set was the concentration of the reporting molecule, Thiazole Orange (TO), a cation whose fluorescence dramatically increases when it binds to nucleic acids, notably highly structured ones (for a review on TO applications in fluorescent sensing, see [9]). The photoactivation of this molecule by flashlightes at a specific wavelength induces the formation of intracellular ROS [1]. Then, ROS destructive action leads to a perturbation of cell homeostasis that could impact cation transporters responsible for TO constant export outside of cells. Consequently, more TO molecules are sequestered in the cell and bound to structured nucleic acids, resulting in a progressive and strong fluorescence increase. Thus, determining the optimal TO concentration is important to obtain the best cellular response to a light flash. Ensuring that the dosage of TO employed does not trigger any toxicity to the cell is equally crucial. We tested four different TO concentrations in the reaction mix: 128, 64, 32 and 16 µM (Figure 2). For 64 and 32 µM, we observed the expected pattern: a stabilization of the fluorescence signal during the 30 min of pre-incubation and a progressive increase in fluorescence in response to successive illuminations, with the best Fpost/Fpre-illumination plateau ratio obtained with 64 µM of TO (see Figure 2). At a concentration of 16 µM, we did not detect any response to the illumination; the amount of TO is probably too low to produce enough ROS to ensure the cellular effect inducing the fluorescence augmentation. At 128 µM of TO, a biphasic effect is observed: a first increase rapidly followed by an unexpected sharp decrease in fluorescence. This atypical profile has been already observed in human cell lines treated with high concentrations of TO (>128 µM) (unpublished data). To date, it remains unexplained. Based on these results, the TO concentration of 64 µM was chosen as the optimal parameter for the further experiments.

#### 2.1.3. Determining the Optimal Growth Phase of the Yeast Culture

The morphology, metabolism and nucleic acid content of the yeast cell change throughout the cell cycle. There are also significant variations in the stress sensitivity between cells at different times of the growth [10,11,12]. The reasons explaining this phenomenon are the changes in cell wall thickness and composition, as well as the activation or repression of specific pathways. We noticed that the AOPY response level of the same strain also showed considerable fluctuation depending on the culture growth phase, attesting to the importance of adjusting this key parameter for the test. Frozen aliquots of wild-type (wt) yeast strain BY4741 were inoculated at OD_600_ = 0.05 in a YPD medium (see Section 4), and samples taken at different times of growth were used to trace the growth curve and to carry on the AOPY tests at these different growth curve points. Such culture, started from frozen cells, showed a lag phase of approximately 10 h long followed by an exponential growth phase starting around 12 h after the inoculation. Finally, cells entered the stationary phase after about 22–23 h of culture time (Figure 3a).

We performed AOPY tests on the samples taken every two hours from the growing culture. The curves of four representative cell growth time points of these experiments are shown in Figure 4. Cells in an early lag phase (0–6 h after the inoculation) showed a high level of TO fluorescence during the installation phase before illuminations. This initial fluorescence level was decreasing in successive samples taken during the lag phase. In addition, these cells did not demonstrate any fluorescence increase after we applied light flashes (see Figure 4a). This could be due to the consequences of freezing on the integrity of yeast cell walls and membranes. In addition, an increase in the intracellular ROS level after a freeze/thaw cycle has been described (for a review, see [13]); we can hypothesize that this oxidative stress could impact the cell’s response to TO treatment by impairing the control of the intracellular level of the biosensor. Closer to the exponential phase, the profiles started to resemble more the classical AOP1 response (see Figure 4, compare panels b and c). Around 12–16 h (exponential growth), we observed the best kinetics: an equilibrium during TO incubation leading to a low and stable fluorescence signal and a progressive fluorescence increase following the illuminations. For cell samples taken along the stationary phase, illuminations resulted in less and less fluorescence increase. For example, we observed only a 2-fold increase for 23 h of growth (Figure 4d) instead of a 15-fold increase during the exponential phase (12 h). Since this parameter, namely the ratio of the Fpost/Fpre-illumination plateau, is the most important for the sensitivity of the test, we concluded that measurements of the antioxidant power of the studied molecules should be carried out on the exponentially growing cells. In our set-up, this corresponds to 12–16 h of growth from the frozen aliquoted cell of the BY 4741 strain (see Figure 4c). 

### 2.2. Testing the AOPY Response to Externally Added or Internally Produced Antioxidants 

After finding the optimal parameters of the AOPY protocol, we used it to test the effect of well-known soluble antioxidant molecules added in the culture medium, as well as of the internal effect of β-carotene synthetized in yeast cells by the expression of a heterologous pathway. 

#### 2.2.1. Testing the Response to Soluble Molecules Added to the Medium 

In order to check the feasibility of the AOPY method and to compare its efficiency to the AOP1 method performed on human cell lines [2], we conducted the assay on wild-type yeast treated with four classical antioxidants. We chose to test three polyphenols with hydrophilic properties, the ability to penetrate cells and free-radical direct scavenging properties: resveratrol (3,4′,5-trihydroxystilbene) belonging to the stilbene family, present in the skin and seeds of red grapes, red wine, peanuts, apples and groundnuts [14]; epigallocatechin gallate (or EGCG, 3,4,5-trihydroxybenzoate de (2*R*,3*R*)-5,7-dihydroxy-2-(3,4,5-trihydroxyphényl)-3,4-dihydro-2*H*-chromén-3-yle), a flavonoid abundant in green tea [15] and quercetin (2-(3,4-Dihydroxyphenyl)-3,5,7-trihydroxy-4H-1-benzopyran-4-one, 3,3′,4′,5,6-pentahydroxyflavone), a second flavonoid present in various vegetables as well as in tea and red wine [16]. The fourth compound, astaxanthin ((3S,3′S)-3,3′-Dihydroxy-β,β-carotene-4,4′-dione), is a lipophilic antioxidant with highly potent peroxyl radical scavenging activities. It belongs to the xanthophyll carotenoid family originally isolated from the lobster *Astacus gammarus* [17].

Yeasts grown in a complete medium were collected in the exponential growth phase and resuspended at OD_600_ 0.6 in a synthetic minimal medium without glucose. The tested antioxidants were then applied at nine increasing doses for 1 h prior to the AOPY revelation procedure. As shown in Figure 5a, increasing doses of resveratrol, quercetin or EGCG delayed or even abolished the increase in fluorescence level induced by illumination flashes. This change in the fluorescence profiles is entirely consistent with the results obtained with the AOP1 method on human hepatocytes [2], and it reveals the capacity of these molecules to penetrate into cells and neutralize the intracellular ROSs. On the contrary, astaxanthin, which cannot cross the plasma membrane, has no impact on the fluorescent signal at any tested dose. This lack of effect is once again in agreement with the results obtained on human hepatocytes for which only a weak partial effect could be observed [2]. The AOPY assay, as the AOP1, specifically reveals the intracellular antioxidant effect of compounds. 

Ratios were calculated for each tested condition using the fluorescence signal after several light flashes (Fpost) and the signal recorded before the first light flash (Fpre). Fpost corresponds to the fluorescence observed at the light flash number required to achieve maximum fluorescence in the control condition. Then, the ratios were used to establish dose–response profiles and calculate, when possible, the half-maximal effective concentration (EC_50_) (Figure 5b, see Section 4 for the details of the calculation). The EC_50_ and coefficient of determination are summarized in Table 1. The three polyphenols have EC_50_ values close to those obtained on hepatocytes after 4 h of treatment [2]. For example, EGCG has an EC_50_ of 7.09 µM with AOP1 in HepG2 cells and 8.962 µM with AOPY on *S. cerevisiae* BY4741. The effect of resveratrol and quercetin in yeast is slightly higher than the effect on hepatocytes (quercetin 23.66 µM and resveratrol 64.66 µM). Taken together, these data confirm that the AOP1 method can be adapted to yeasts to test the intracellular antioxidant effect of hydrophilic molecules added directly to the culture medium. 

#### 2.2.2. Testing the Response to Internally Produced Antioxidant Molecules

Finally, we attempted to use our method for the intracellular detection and quantification of carotenoids, which are characterized as excellent lipophilic antioxidants [18,19,20]. This step was intended as a proof of concept for monitoring antioxidant compounds obtained by synthetic biology, directly in the producing microorganisms, as the main interest of this approach is that it would allow avoiding the harvesting, cell lysis and analytical quantification steps of the synthetic biology processes. Yeasts, including *Saccharomyces cerevisiae*, are one of the choice hosts for this heterologous production [21,22,23,24,25,26]. For this purpose, we first compared the viability after the TO and illumination treatment in the same conditions as the AOPY protocol of yeast strains producing different antioxidant molecules [25] to the original host strain, BY4741a, below referred to as “Control”. 

The viability test results shown in Figure 6 seem to confirm that the antioxidant molecules produced inside the cells protect them from dying upon TO and illumination treatment. It is worth mentioning that incubation with TO without light activation does not cause a viability decrease. Hence, the protective effect seen in Figure 6 is most likely achieved by the carotenoids scavenging and neutralizing the ROS formed by the TO light activation. The next step was to check the responses of those different strains in the AOPY test conditions.

Even if this growth phase is not the optimal one, strains were grown almost until the stationary phase in order to ensure a good accumulation of carotenoids inside the cells [21]. We obtain an Fpost/Fpre-illumination fluorescence ratio of 6.53 with the control strain (BY4741) and a complete absence of response for all three producing strains (Figure 7). One of the most probable interpretations of this result is the antioxidant activity of the intracellular carotenoid molecules produced protecting the cells against the induced ROS, as recently reviewed [27].

#### 2.2.3. Quantitative Detection of Intracellularly Produced Carotenoids

Considering that our test could be of great use for the various experimental strategies aiming at improving carotenoid production in yeast strains, we wished to assess the quantitative or semi-quantitative aspect of the effect observed above. Hence, the next challenge was to see if the AOPY assay would discriminate between different concentrations of intracellular carotenoids. We collected the cells at different time points of the growth curve and tested AOPY response with parallel measurement of the β-carotene amount per cell. The BY4741 host strain was used as a control to separate the impact of the growth phase on the AOPY response (described in Section 2.1.3) to the specific effect of β-carotene. For each analyzed point, the growth phases of the control strain BY4741 and the β-carotene-producing strain were comparable. Hence, we consider it legitimate to compare the AOPY results obtained at each time point for these two strain samples.

Starting from 15 h of growth, four time points were analyzed. For all, the increase in TO fluorescence was lowered in the strain producing β-carotene, becoming almost null at 24 h, while the BY4741 control strain was still reacting at this time. This is clearly visible when comparing the AOPY response (Fpost/Fpre ratio) of the two strains at successive time points of the culture (Figure 8a). The relative effect of the β-carotene production at each growth point was calculated as the ratio between the maximal AOPY response (Fpost/Fpre ratio) of the β-carotene strain to the one of the BY4741 control strain (Figure 8b, upper panel). In parallel, the amount of β-carotene produced at the different time points was quantified using a spectrometric method (OD_480 nm_ after acetone extraction). (Figure 8b, lower panel). We could see a good correlation between the AOPY response and the β-carotene amount per cell, confirming that the AOPY test could be used for a rapid and easy screening of the intracellular production of β-carotene in yeast cells. 

## 3. Discussion

This work describes the development of a new assay for monitoring the antioxidant power of compounds either added externally or produced internally, in live yeast cells. As one of the best genetically developed eukaryotic models, with a high growth rate and very simple nutritional requirements, the baker’s yeast *S. cerevisiae* presents several key advantages to design biosensors [5,6]. Hence, we chose this cellular model to design a microbial cell-based bioassay for the analysis of antioxidants, based on the LUCS technology of ROS induction [1]. We first determined the optimal parameters for performing an antioxidant detection assay on live yeast cells. Several parameters such as the growth medium, growth phase of the yeast culture, cell density, buffer, TO concentration and the intensity, number and duration of the light pulses were optimized. Then, we validated this method using antioxidants externally added on the yeast cells. Finally, we attempted the detection and quantification of carotenoids, which are strong liposoluble antioxidants, produced in yeast strains. This last step was intended as a proof of concept for the ability of LUCS technology to monitor antioxidant compounds produced by synthetic biology directly in the producing microorganisms. Here, we demonstrate that this assay could be used to monitor the amount of antioxidant compounds produced intracellularly by the yeasts expressing a synthetical metabolic pathway.

Regarding the AOPY assay, developing it on yeast cells is already interesting. Indeed, the use of cells that are easy and cheap to grow, keep and transport opens the possibility for testing the pro- or antioxidant properties of samples without the need for sophisticated laboratory equipment mandatory for higher eukaryotic cell culture manipulations. In addition, yeasts’ well-developed genetic tools provide the possibility to further modulate the sensitivity of the test: yeast mutant libraries can be used, for example, to select cell walls or membrane mutants better adapted for specific conditions or for testing various types of molecules. Moreover, yeasts and especially the budding yeast *S. cerevisiae* are very often used as eukaryotic cell models to study stress response, aging mechanisms and specific cellular effects of pro- and antioxidant molecules. In these studies, several different tests are used to assess either the oxidative stress level or the antioxidant power of molecules added to fight this stress (see for examples [28,29,30,31]). The most commonly used method is exposing yeast cells to a strong oxidant agent such as H_2_O_2_, and the most classical read-out for their response is a simple viability test: yeast cells are plated on a solid-agar-rich growth medium and incubated at 28 °C for 72 h, and finally colonies formed are counted. Other methods involve dichloro-dihydro-fluorescein diacetate (H2DCFDA) [32] which is the base of the cellular antioxidant activity (CAA) assay [33]. Transcriptional reporters have been used as well, notably based on the Yap1 transcription factor [34] but also on well-known yeast stress response genes such as *MSN2* and *MSN4* [35], and a method based on cellular growth arrest has been described [36]. Compared to these existing antioxidant detection methods, the AOPY assay developed in this study presents several advantages. First, it is much faster as it does not require any growth time delay and no colony counting on plates. In addition, no further steps for stress level detection after the ROS induction are needed, neither is any tagging protein modification (e.g., YFP) for microscopy such as with the Yap1-based reporter, or staining prior cytometry analysis used for aging and viability. Moreover, in the AOPY assay, the ROS production by TO photoinduction is directly intracellular, fast and simultaneous for 96 samples or more, depending on the microwell plate and illuminator choice. Signal detection is also more informative compared to the other methods classically used, since AOPY provides not a single value but full fluorescence curves, which allow for having a fine quantitative response correlating with the amount of the tested molecule as well as a better qualitative insight on the cell’s response dynamics and fitness. Finally, regarding equipment requirements, AOPY only requires a small and inexpensive bench LED illuminator and a fluorescence plate reader which is now becoming a common piece of lab equipment.

However, the main achievement of this study is likely the potential role of the AOPY assay developed here for the synthetic biology field. Indeed, the world demand for natural antioxidants is very high, particularly for the human nutrition, health (both pharmaceutical and global well-being) and cosmetic industrial domains. The difficulties and costs of isolating them from natural sources have led to a worldwide competition for the design of alternative production strategies in microorganisms, with a need to follow their accumulation during the processes. Hence, the capacity to monitor the biosynthesis of the product of interest directly within the producing microorganisms (e.g., bacteria, yeasts, micro-algae, etc.) would be a key advantage. We showed here that the response level of the AOPY assay depends on the intracellular amount of the antioxidant molecule produced, which indicates that it could be used as a biosensor-type tool to follow the production level in vivo. Since a great number of molecular engineering pathways have been developed in *Saccharomyces cerevisiae* to produce various compounds with antioxidant properties such as squalene [37], astaxanthin [23], violaxanthin [38] kampferol [39], α-humulene, δ-guaiene, α-santalene, β-eudesmol [40] and many more, we believe that the AOPY assay is of interest for the industrial use and potential further improvements in these yeast strains.

Finally, our work opens the possibility to study the effects of heterologous molecules within the model microorganism used for their production, which might represent a great interest for the fundamental physiological studies of the changes in cell homeostasis induced by the synthetical pathway expression. 

## 4. Materials and Methods

### 4.1. Yeast Strains Used in This Study

All strains used in this study are in the BY4741 background (MATa *his3Δ1 leu2Δ0 met15Δ0 ura3Δ0*), deleted for the *GAL80* gene (*gal80:MET15*) to release the glucose catabolic repression of the gal promoters (*GAL1,10*) controlling the genes of the carotenoid pathway.

The carotenoid-producing strains all had integrated cassette with CrtE *GAL1-10*-t*HMG1* to keep high GGPP level, with the following specific additions:

“Phytoene” strain had integrated *CrtB* gene (yHR21 *Δho::CrtB-Gal1-10*).

“Lycopene” strain had integrated *CrtB* and *CrtI* genes (yHR21 *Δho::CrtB-Gal1-10 ΔTy4:: CrtI-Gal1-10*).

“β-Carotene” strain had integrated *CrtB, CrtI* and *CrtY* genes (yHR21 *Δho::CrtY(yb)B(ek)I-Gal1-10*).

All construction details are described in [21].

### 4.2. Yeast AOP Assay Experimental Protocol

Yeast cells were cultivated in liquid YPD medium at 30 °C under agitation. For AOPY assay, cells were centrifuged and resuspended in YNB medium without carbon source at desired OD_600_. Fluorescence values of TO were measured using fluorescence readers CLARIOstar^®^Plus™ (BMG-Labtech, Champigny s/Marne, France; Figure 4 and Figure 8) or Varioskan™ LUX (Thermo Fisher Scientific Waltham, MA, US; Figure 1, Figure 2, Figure 3, Figure 5 and Figure 7) at the 505/535 excitation/emission setting.

#### 4.2.1. AOPY Protocol Optimization

For each test, 90 μL of these cells was mixed with 10 μL of TO at 640 μM to get the final TO and cell concentrations of 64 μM and 0.6 (OD_600_), respectively. Experiments were performed in 96-well microplates with technical triplicate. During the first phase of the test, which serves for the equilibration of TO concentration inside the cells, measurements were made every 5 min for 30 min. After the equilibration step, light treatment of the samples was performed by applying 15–20 light flashes with a 96-well plate illuminator with fixed parameters (96 LEDs, 470 nm, 10 s, 24 mW/cm^2^). Every flash is followed by a measurement of TO fluorescence. When the fluorescence meanings stopped increasing after few successive light flashes, we considered that the post-illumination stabilization of fluorescence was achieved. To evaluate the response efficiency, we calculated the ratio of stabilized fluorescence after and before the light treatment = Fpost/Fpre-illumination ratio.

#### 4.2.2. Test of Antioxidant Molecules with AOPY Protocol

Stock solutions of Quercetin (Q4951, Sigma-Aldrich, St. Louis, MO, US), Resveratrol (R5010, Sigma-Aldrich), Epigallocatechin Gallate (E4143 Sigma-Aldrich) and Astaxanthin (SML0982, Sigma-Aldrich) were prepared in advance in 100% DMSO or 100% EtOH, aliquoted and stored at −20 °C. For the dose–response experiments, nine different concentrations were obtained by serial factor 2 dilutions. Solvent was maintained at 1% (*vol/vol*) in all the solutions tested. Experiments were carried out in 96-well microplates without using the edge wells. Each experimental condition was assayed in triplicates, including the solvent control without sample (1% DMSO or EtOH in YNB medium).

Cells in YNB medium at final OD_600_ were first incubated with antioxidant molecules for 1 h at 30 °C under shaking. TO prepared in YNB medium was added to the cells for 40 min at the final concentration of 64 µM. During this TO incubation, the fluorescence (505/535 nm) was measured every 5 min in Varioskan™ LUX (Thermo Fisher Scientific). After the stabilization of TO, microplate was placed in a dedicated illuminator (24 LEDs, 470 nm, each LED centered on the intersection of 4 wells) (provided by LED Engineering Development, Montauban, France) and illuminated at 24 mJ/cm^2^. Fluorescence level was measured immediately after illumination (flash number 1). The same illumination/reading cycle was repeated at least 13 times.

Dose–response curves were measured by Fpost-illumination/Fpre-illumination fluorescence ratio. Then, to calculate 50% efficacy concentration (EC_50_) values, we used a mathematical non-linear regression model (sigmoid fit) given by Prism9 (Graph pad), that follows the following equation: Y = Bottom + (Top − Bottom)/(1 + 10^(LogEC50 − X)×HillSlope^), where HillSlope = slope coefficient of the tangent at the inflection point. EC_50_ and R^2^ values were deduced from the regression model.

### 4.3. UV Detection of Carotenoids

Based on the optimized protocol described by [41], after 15, 18, 21 and 24 h of culture in YPD medium at 30 °C, cells were centrifuged at 6000× *g* for 10 min, and pellets were lyophilized and weighed. Lyophilized pellets were then dissolved in acetone and broken by 3 cycles of 20 s in a Fast Prep FP120 (Thermofisher, Waltham, MA, USA). The absorbance at 480 nm of cleared acetone extracts was measured as a proxy for carotenoid level as described by [42,43] and normalized by the dry cell weight of each sample.

### 4.4. Viability Tests

After an AOPY assay, untreated (without TO) and TO-treated cells were reserved for viability tests. Samples of 1 μL were taken after the 15 light illumination flashes, spread on YPD plates and cultured for 32 h at 30 °C. Grown colonies were counted for each sample. We calculated a percentage of the grown colony number for TO-treated cells by comparison with the non-treated-cell-grown colonies considered equal to 100%.

## 5. Conclusions

In conclusion, our study presents the development of a new method for antioxidant power detection using live yeast cells. Easy and rapid, this assay provides quantitative data (EC_50_) demonstrating the capacity of compounds to neutralize free radicals and ROS produced by a cell under controlled conditions. It is, hence, a fine tool for the estimation of the amount of antioxidants added to the cell medium, as we demonstrated here on several well-known antioxidant molecules. In addition, the remarkable potential of this method is the possibility to measure, characterize and study the physiological effect of antioxidant molecules produced inside the cells of natural or genetically modified microorganisms. These studies may be useful in synthetic biology for improving producer strains, as well as in fundamental research for better understanding oxidative stress response mechanisms.

## Figures and Tables

**Figure 1 ijms-24-11800-f001:**
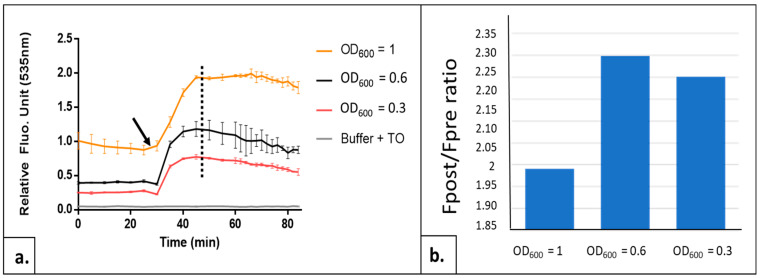
Influence of cell density on AOPY test sensitivity. (**a**) Kinetics of TO fluorescence obtained with AOPY test on BY 4741 yeast strain collected in exponential growth phase and diluted at different cell densities (measured by optical density at 600 nm). Yellow curve: OD_600_ = 1, black curve: OD_600_ = 0.6 and orange curve: OD_600_ = 0.3; gray curve = control without cells (buffer + TO). Black arrow indicates the starting point of illuminations. From this point, each illumination is followed by a fluorescence measurement. Data are the mean of 3 independent experiments, with each sample in 3 replicates. (**b**) Fpost/Fpre-illumination ratios for these same cell densities. Fpost = fluorescence value at flash number indicated as dotted line on graph a (corresponding to the highest control value). Fpre = fluorescence value before the first light flash.

**Figure 2 ijms-24-11800-f002:**
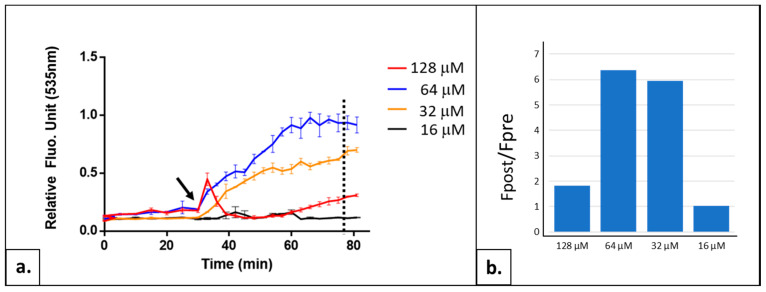
AOPY profiles and Fpost/Fpre-illumination ratios for different TO concentrations. (**a**) Kinetics of TO fluorescence of BY4741 cells collected in exponential growth phase, diluted at OD_600_ 0.6 and treated with 16, 32, 64 or 128 μM of TO. Black arrow: starting point of illumination. Experiments were carried out twice in triplicate with similar results. (**b**) Ratio of Fpost/Fpre-illumination plateau is compared for these four TO concentrations. Fpost = fluorescence value at flash number indicated as dotted line on graph A (corresponding to the highest control value). Fpre = fluorescence value before the first light flash. Data are the mean of two independent experiments, with each sample in triplicate.

**Figure 3 ijms-24-11800-f003:**
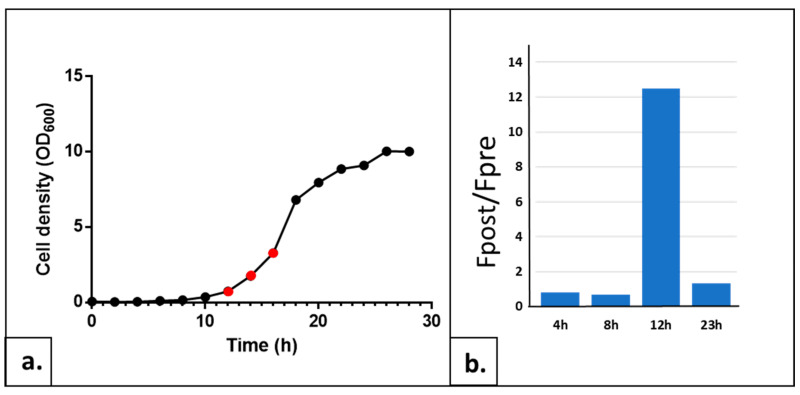
Influence of the growth phase of yeast cells on AOPY test sensitivity. (**a**) Growth curve of BY4741 wt strain started in YPD medium from the −80° frozen cell aliquots established by measuring the optical density at 600 nm every 2 h for a total of 28 h. Red points indicate the culture time with the best AOP response (see below). (**b**) Fpost/Fpre-illumination ratios of the AOPY test at four chosen time points of the culture representative of the lag phases (4 h and 8 h) and exponential growth phases (12 h) and (23 h).

**Figure 4 ijms-24-11800-f004:**
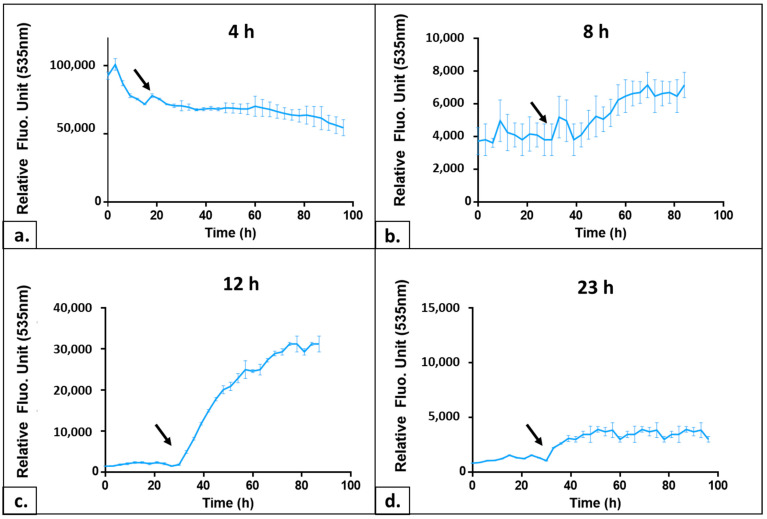
Influence of the growth phase on AOPY fluorescence profiles. Kinetics of TO fluorescence in AOPY assay using cells collected in different phases of the cell growth curve (see Figure 3): (**a**) after 4 h of culture, i.e., lag phase; (**b**) after 8 h, end of lag phase; (**c**) after 12 h, exponential growth; (**d**) after 23 h*,* i.e., stationary phase. Black arrow: starting point of illumination. RFU scales are different for a better adaptation to each profile.

**Figure 5 ijms-24-11800-f005:**
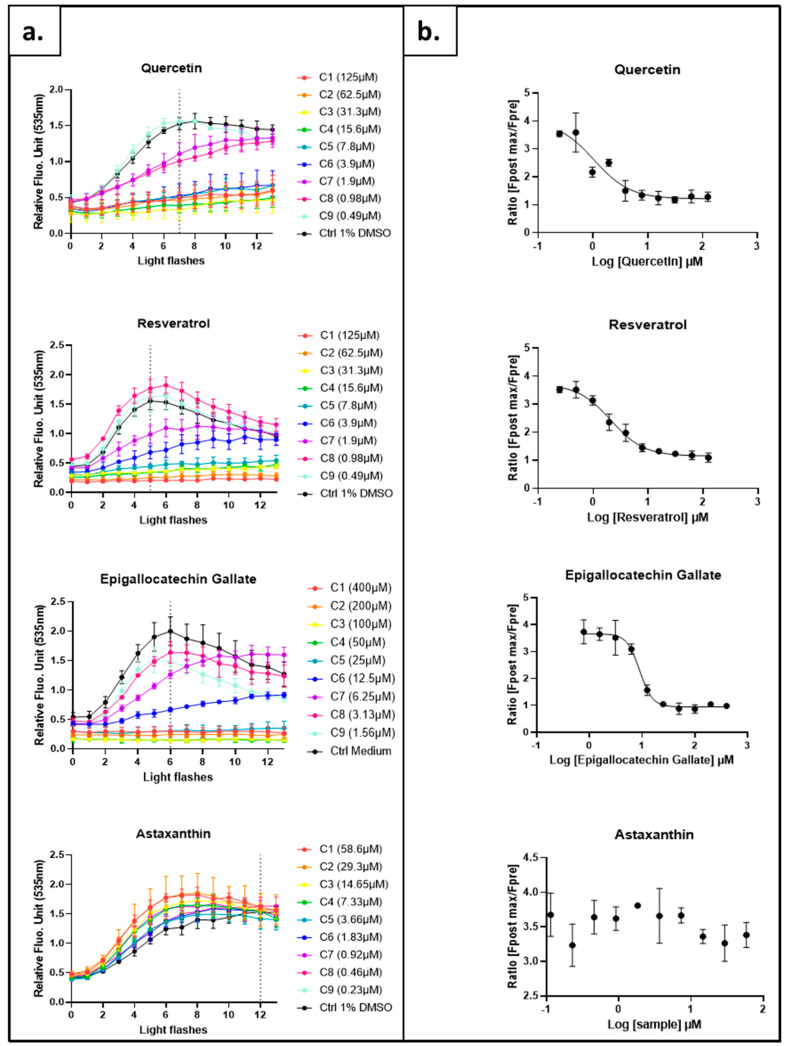
Comparison of standard antioxidant profiles and dose–response curves by AOPY in BY4741 strain. (**a**) Kinetics of fluorescence obtained in exponential growth phase (19 h) for each tested concentration using optimized light application (24 mJ/cm^2^), TO concentration (64 µM) and OD_590 nm_ (0.6). (**b**) Dose–response curves measured by fluorescence ratio. Fpost = fluorescence value at flash number indicated as dotted line on graph A (corresponding to the highest control value). Fpre = fluorescence value before the first light flash. All experiments were carried out twice in triplicate with similar results.

**Figure 6 ijms-24-11800-f006:**
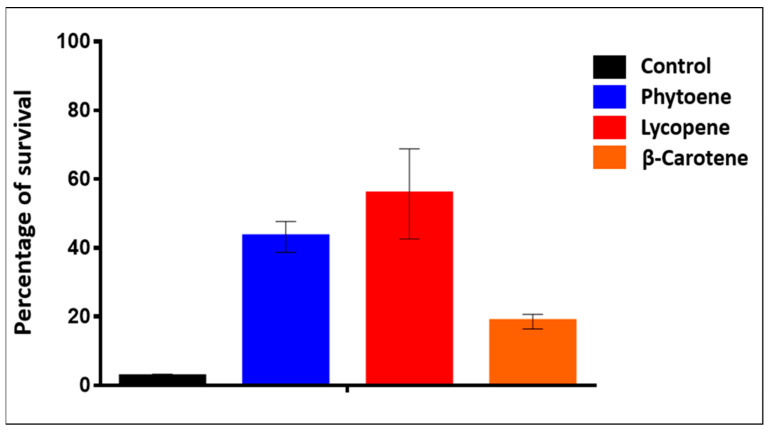
Viability test of yeast cells upon AOPY assay. Histograms represent the percentage of yeast cell survival (able to form colonies when plated on complete solid medium; see Section 4) of four yeast strains after the incubation with Thiazole Orange and illumination flash sequence of AOPY assay. Black = Control strain BY4741, Blue = Phytoene-producing strain, Red = Lycopene-producing strain, Orange = β-carotene-producing strain.

**Figure 7 ijms-24-11800-f007:**
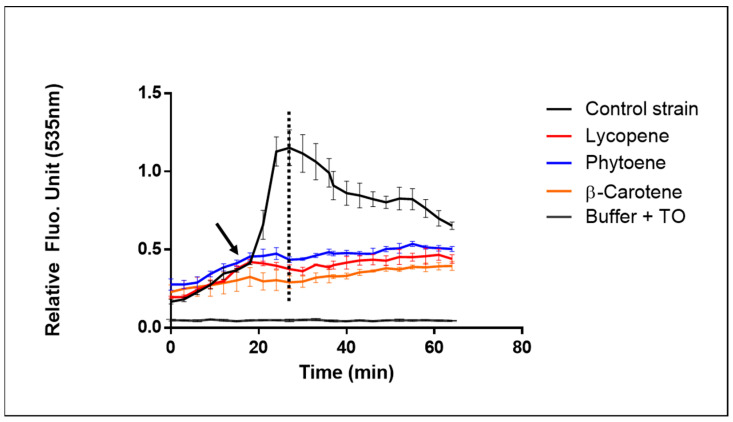
Effect of intracellularly produced carotenoids on AOPY response. AOPY assay was conducted on three different carotenoid-producing yeast strains: black curve = Control strain BY4741 blue curve = Phytoene-producing yeast strain, red curve = Lycopene-producing strain, orange curve = β-carotene-producing yeast strain, gray curve = no cells (buffer + TO). Black arrow indicates the starting point of illumination. Dotted line indicates the highest control value used to calculate Fpost/Fpre-illumination fluorescence ratio. Experiment was carried out three times with similar results, with each sample in triplicate.

**Figure 8 ijms-24-11800-f008:**
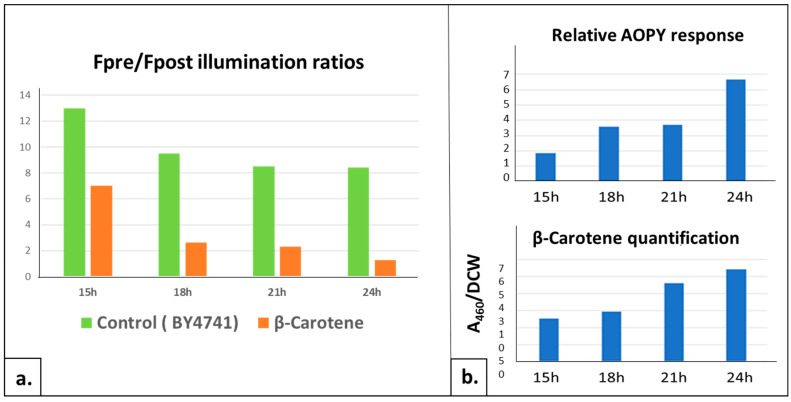
Correlation of intracellularly produced β-Carotene amount and AOPY response. β-Carotene-producing yeast strain and control strain (BY4741) were submitted to AOPY test and parallel β-carotene quantification at different growth time points. (**a**) AOPY signals (Fpost/Fpre-illumination ratios) at 15 h, 18 h, 21 h, 24 h of BY4741 (green columns) and β-carotene-producing (orange columns) strains. (**b**) Relative effect: ratios between AOPY response level (Fpost/Fpre-illumination ratios) of β-carotene-producing strain and control strain BY4741 for each time point are represented as histograms (**upper panel**). Quantification of β-carotene by UV absorbance at 480 nm reported to corresponding dry cell weight (DCW) (**lower panel**).

**Table 1 ijms-24-11800-t001:** EC50, CI 95% and R^2^ values obtained from the dose–response curves depicted in Figure 5b. ND: not determined.

Compound	EC50 (µM)	CI 95% (µM)	R^2^
Quercetin	1.041	0.02520 to 1.961	0.86
Resveratrol	2.086	1.529 to 2.680	0.97
Epigallocatechin Gallate	8.962	7.711 to 10.40	0.96
Astaxanthin	ND	ND	ND

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
