# Peer review of "A New Role for Yeast Cells in Health and Nutrition: Antioxidant Power Assessment"

_ijms, 2023, doi:10.3390/ijms241411800_

Round 1

Reviewer 1 Report

-keywords should be rearranged in alphabetical order

The abstract part should be a mini article, that is, the information presented in the article draft should be written briefly, emphasizing important details

- The results in the discussion section should be discussed in the light of previous studies, deficiencies should be criticized, positive results should be highlighted and suggestions should be made in the light of these results

- References such a study is very few, most of it belongs to materials and methods

-Which protocol was used in the preparation of the yeast extract, should be cited with reference.

-The article should be revised according to the journal writing rules.

-Purpose should be clearly stated

-introductory part is enough

Author Response

Answers to the Reviewers

The authors are grateful towards the two reviewers for their careful reading of the manuscript and their pertinent comments and suggestions which have greatly helped us to improve the manuscript. All in one, the purpose of this work is now much clearer from the beginning of the article and the novelty of our work better emphasized.  Figures have been improved, the discussion significantly enriched and the conclusions stated separately. Below are the responses to the specific points addressed by the reviewers.

Review 1 Report Form

Comments and Suggestions for Authors:

keywords should be rearranged in alphabetical order

Thank you for noticing this, we rearranged them.

The abstract part should be a mini article, that is, the information presented in the article draft should be written briefly, emphasizing important details

We modified the abstract accordingly, and also took into account the request of reviewer 2 to provide background already in the abstract. 

The results in the discussion section should be discussed in the light of previous studies, deficiencies should be criticized, positive results should be highlighted and suggestions should be made in the light of these results

We modified the discussion part and notably added a significant new paragraph describing the previously existing methods and discussing our assay’s differences, advantages and potential difficulties (such as equipment required) compared to them.

References such a study is very few, most of it belongs to materials and methods

Four new references were added for background in the introduction part, and nine new ones were introduced and commented in the discussion part.

Which protocol was used in the preparation of the yeast extract, should be cited with reference

Does reviewer 1 refer to the UV detection of carotenoids produced by the yeast strains? If yes, we did our best to accurately describe the optimized process that we used, and cited the three relevant references in this entry of the material and methods section.

The article should be revised according to the journal writing rules.

We attempted to indeed revise the article accordingly.

Purpose should be clearly stated

It is now much more clearly stated, already in the abstract, but also in the introduction, discussion and conclusion.

introductory part is enough

Reviewer 2 Report

What is the novelty and originality of this work? Which should be clarified in the introduction

The abstract section lacks background and a gap

The conclusion section should be added

The quality of figures should be improved

figure 8 and its title should be improved

Author Response

The authors are grateful towards the reviewers for their careful reading of the manuscript and their pertinent comments and suggestions which have greatly helped us to improve the manuscript. All in one, the purpose of this work is now much clearer from the beginning of the article and the novelty of our work better emphasized.  Figures have been improved, the discussion significantly enriched and the conclusions stated separately. Below are the responses to the specific points addressed by the reviewers.

Review 2 Report Form

Comments and Suggestions for Authors

What is the novelty and originality of this work? Which should be clarified in the introduction

              This work describes the development of a new assay for antioxidant power detection, based on the yeast S. cerevisiae as cell model and using TO photoinduction to produce ROS intracellularly. This is now clearly stated in the introduction.

The abstract section lacks background and a gap

              We added a sentence at the beginning of the abstract to provide background and corrected it to strengthen the novelty aspect (gap) of our work.   

The conclusion section should be added

              It is added now.

The quality of figures should be improved, figure 8 and its title should be improved

              We improved the quality of Figures, notably Figure 4 in which the two bottom graphs c and d whose curves were previously not as visible as the others upon pdf conversion. They appear now all four as visible.

In addition, Figure 8 has also been significantly modified. The four curve graphs of the panel a. of this figure have been replaced by coloured histograms. We believe that the new Figure 8   is much nicer and that the important information of this figure is now easier to get.

Round 2

Reviewer 1 Report

Congratulations

Reviewer 2 Report

the authors responded appropriately to each suggestion